# A Multi-Enzyme Cascade Response for the Colorimetric Recognition of Organophosphorus Pesticides Utilizing Core-Shell Pd@Pt Nanoparticles with High Peroxidase-like Activity

**DOI:** 10.3390/foods12173319

**Published:** 2023-09-04

**Authors:** Zainabu Majid, Qi Zhang, Zhansen Yang, Huilian Che, Nan Cheng

**Affiliations:** 1Beijing Laboratory for Food Quality and Safety, College of Food Science and Nutritional Engineering, China Agricultural University, Beijing 100083, China; zaymajid88@gmail.com (Z.M.); zhangqi20182154@163.com (Q.Z.); haixinluohua@163.com (Z.Y.); chehuilian@cau.edu.cn (H.C.); 2Key Laboratory of Precision Nutrition and Food Quality, Key Laboratory of Functional Dairy, Ministry of Education, College of Food Science and Nutritional Engineering, China Agricultural University, Beijing 100083, China

**Keywords:** Pd@Pt nanoparticles, organophosphorus pesticides, acetylcholinesterase, colorimetric detection, multi-enzyme cascade

## Abstract

In modern agricultural practices, organophosphorus pesticides or insecticides (OPs) are regularly used to restrain pests. Their limits are closely monitored since their residual hinders the capability of acetylcholinesterase (AChE) and brings out a threatening accumulation of the neurotransmitter acetylcholine (ACh), which affects human well-being. Therefore, spotting OPs in food and the environment is compulsory to prevent human health. Several techniques are available to identify OPs but encounter shortcomings like time-consuming, operating costs, and slow results achievement, which calls for further solutions. Herein, we present a rapid colorimetric sensor for quantifying OPs in foods using TMB as a substrate, a multi-enzyme cascade system, and the synergistic property of core-shell Palladinum@Platinum (Pd@Pt) nanoparticles. The multi-enzyme cascade response framework is a straightforward and effective strategy for OPs recognition and can resolve the previously mentioned concerns. Numerous OPs, including Carbofuran, Malathion, Parathion, Phoxim, Rojor, and Phosmet, were successfully quantified at different concentrations. The cascade method established using Pd@Pt had a simple and easy operation, a lower detection limit range of (1–2.5 ng/mL), and a short detection time of about 50 min. With an R^2^ value of over 0.93, OPs showed a linear range of 10–200 ng/mL, portraying its achievement in quantifying pesticide residue. Lastly, the approach was utilized in food samples and recovered more than 80% of the residual OPs.

## 1. Introduction

The use of organophosphorus pesticides (OPs) in agricultural production has recently increased due to their low cost, simple synthesis, and effectiveness as insecticides. However, their improper use can negatively impact consumable products, the climate, and human well-being. Consuming water and food contaminated with OPs can adversely affect human health by attacking the sensory system and causing illness such as paralysis. These harmful effects are the results of the inhibitory function of acetylcholinesterase (AChE) and an increase ratio of the neurotransmitter acetylcholine (ACh) [1,2,3,4,5,6,7,8]. This makes recognition of OPs in foods and the environment compulsory, and their maximum residue limits are strictly monitored. There are several methods for identifying OPs in foods and the environment such as immunoassays and chromatographic techniques [9,10,11,12,13], but they require skilled personnel, costly materials, and delayed results output, which present limitations for on-site detection [1,13,14,15,16,17]. Thus, making a fast and simple method for distinguishing various OPs in food and climate without delay is direly required. This is why we focused on developing a rapid colorimetric sensor for quantifying OPs in foods using TMB as a substrate, a multi-enzyme cascade system, and the synergistic properties of palladium and platinum core-shell (Pd@Pt) nanoparticles (NPs). The multi-enzyme cascade response is a straightforward and effective technique for OPs recognition and can resolve the mentioned concerns. The mechanism involves incorporating the joint effect of AChE and oxidative choline (ChOX) toward hydrolyzing ACh and conveying H_2_O_2_, while the Pd@Pt catalyzes TMB oxidation. In the presence of OPs, the ability of AChE is obstructed, which results in differential visual instabilities [17]. However, our proposed approach successfully quantified the randomly selected OPs.

Nanozymes are made of materials that can mimic the activities of ordinary enzymes. Nanozymes attention has been raised since Yan et al. discovered their existence in 2007. Until now, they suppressed regular catalysts after understanding their superior features. By simply varying size, morphology, and alignment, many nanozymes have been discovered and characterized into different groups [18,19,20,21,22,23,24,25,26]. Among many groups, noble metal-based NPs, including gold (Au), platinum (Pt), and palladium (Pd), have been highly exposed recently due to their sturdiness. They possess higher synergistic characteristics toward different enzyme-like activities, including peroxidase (POD)-like activity. Their magnificent synergistic impacts are brought about by the size of NPs (less than 100 nm) and multiple active sites [27,28].

Noble metal-based nanozymes can be made of single or double metallic alloys, which cooperate to enhance and upgrade product catalytic activity. Compared with single-metallic compounds, double-metallic NPs have revealed higher enzyme-like activities. A mixture of core-shell NPs is an engaging type of nanozymes with binary sections, significant molecule size, and multiple active sites, which makes the compounds better catalysts. These alloys are reported to have the best metallic connections due to the pleasant joint efforts of individual elements toward redox responses [27,28,29,30,31,32,33,34,35,36]. Apart from that, the noble metal-based NPs are better electrocatalysts, whereby Pd is the most available metal with suitable substitute features, which makes it a more synergistic enhancer. This factor resulted in the elevation of studies featuring Pd NPs to detect numerous substances [31,32,33,34,35,36,37,38,39,40,41].

The position of the metal and innate nanomaterial properties in nanozymes assists them in maintaining catalytic function even in the worst environments, which makes them more stable and a better substitute for typical enzymes [30,31,32,33,34,35,36,37]. Due to these benefits, nanozymes have exhibited excellent performance in various applications, such as the breakdown of various organic pollutants and the elimination of multi-drug resistant bacteria. Nanozymes have been utilized recently in the progression of brief processes for uncovering various hazardous materials. To date, ions, molecules, and organic compounds can all be qualitatively and quantitatively detected by nanozymes [31,32,33,34,35,36,37,38,39,40,41,42,43,44,45,46,47,48,49], making them the center of attention in bio-sensing fields.

## 2. Materials and Methods

### 2.1. Materials

Potassium tetrachloroplatinum (II.) (K_2_PtCl_4_, 99.9%), tetrachloroauric acid (HAuCl_4_, 99%), ascorbic acid (AA, 99%), hydrogen peroxide (H_2_O_2_, 30%), 3,3’,5,5’-tetramethylbenzidine (TMB, 99%), and Pluronic F127 were purchased from Sigma–Aldrich (St. Louis, MO, USA). In addition, sodium tetrachloropalladate (Na_2_PdCl_4_, 98%) was purchased from Shanghai Maclin Biochemical Technology Co., Ltd. (Shanghai, China). Dimethyl sulfoxide (DMSO, 99.9%) was purchased from Shanghai Aladdin Biochemical Technology Co., Ltd. (Shanghai, China), while HCl and citric acid (99.5) were from the Beijing Chemical Plant (Beijing, China). All reagents used were analytically pure.

### 2.2. Methods

#### 2.2.1. Nanozyme Synthesis

(1) Synthesis of Pd@Pt: 20 mg of Pluronic F127 was dissolved in an aqueous solution containing 1.8 mL, 0.2 mL, and 44 μL of K_2_PtCl_4_ (20 mM), Na_2_PdCl_4_ (20 mM), and hydrochloric acid (6 M), respectively. Then, 2.0 mL of 100 mM of ascorbic acid (AA) was added as a reducing agent. The mixture was continuously sonicated for 4 h. During the ultra-sonication, the temperature was kept constant at 35 °C. Finally, the final product was collected and washed with acetone and water in consecutive washing/centrifugation cycles (10,000 rpm, 5 min) five times and then dried at room temperature [34].

(2) Synthesis of Au@Pd: After dissolving Brij 58 (0.04 g) in 50 mM Na_2_PdCl_4_ and 50 mM HAuCl_4_ aqueous solutions, 2.5 mL of water was added. Then, 1.5 mL of ascorbic acid (0.25 M) was subsequently added as a reducing agent, and the product was mixed and sonicated for 6 h at room temperature. Lastly, the final product was collected and washed with acetone and water in consecutive washing/centrifugation cycles (10,000 rpm, 5 min) five times and then dried at room temperature.

#### 2.2.2. Peroxidase Activity Test for Nanozymes

A 32 μL of 30% bottle (wt/vol) of H_2_O_2_, 32 μL of 20 mM TMB solution, and different volumes of 100-fold diluted nanozyme solution were dissolved in acetic acid buffer at pH 4.6 to prepare 200 μL of solution. The nanomaterial fractions were 0, 1, 2, 4, 8, 16, and 32 μL, respectively. The difference in absorbance (652 nm) after adding nanozymes was recorded every 10 s for 400 s. A nanoparticle with higher POD-like activity was screened for subsequent experiments.

#### 2.2.3. POD Kinetic Test of Nanozymes

Au@Pd and Pd@Pt nanoparticles prepared in Section 2.2.1 were sonicated for 1 h to homogeneous dispersion and then diluted 100-fold. Thereafter, 10 μL Au@Pd (0.006 mg/mL) and Pd@Pt (0.064 mg/mL), 32 μL of 30% H_2_O_2_, and different volumes of 20 mM of TMB solution were dissolved in acetic acid buffer at pH 4.6 and prepared a 200 μL reaction system. The difference in absorbance (652 nm) after adding nanozymes was detected for 60 s before and after the reaction, with TMB concentrations of 0.025, 0.05, 0.1, 0.2, 0.4, 0.8, 1.6, and 3.2 mM, respectively. The OriginPro 2019b software was applied to fit the Michael equation to obtain two key parameters: the Michael constant (*K_m_*) and the maximum velocity (*V_max_*) of the reaction to analyze the POD-like activity of the nanozymes [50].

#### 2.2.4. Effect of pH on Nanozymes Activity

After conducting Section 2.2.2 and Section 2.2.3, a nanozyme with higher POD-like activity was screened for subsequent experiments. Then, 32 μL of 30% H_2_O_2_, 32 μL of 20 mM TMB solution, and 16 μL of 100-fold diluted nanozyme solution with the best catalytic effect, were dissolved in acetate buffer pH 2.6, 3.6, 4.6, 5.6, 6.6, 7.6, and 8.6, respectively. The difference in absorbance (652 nm) after adding the nanozymes was detected and the influence of pH was analyzed. 

#### 2.2.5. Effect of pH on Nanozymes Activity 

A 200 μL solution containing 32 μL of 30% H_2_O_2_, 32 μL of TMB (20 mM), 16 μL of the nanoparticles with higher POD activity, and 120 μL of acetate buffer with pH 4.6 was arranged for analysis at different temperatures. The difference in absorbance (652 nm) after adding nanozymes at 4, 25, 35, 45, 55, and 65 °C was detected and the influence of temperature was analyzed. 

### 2.3. Discrimination of OPs 

Carbofuran (Ca), Malathion (Ma), Parathion (Pa), Phoxim (Ph), Rojor (Ro), and Phosmet (Pho) OPs were picked randomly for quantification. An incubation of 10 min was provided after the addition of 65 μL ACh (7 mg/mL), OPs 10 μL (100 ng/mL), and 25 μL AChE (1 U/mL) in 200 μL PBS (0.1 M, pH 7.4). Then, 20 μL ChOX (5 U/mL) was added, and the response time was 20 min before the addition of 580 μL HAc-NaAc buffer (0.05 M, pH 4.6), 50 μL TMB (20 mM), and 50 μL Pd@Pt nanozymes (0.05 mg/mL). Thereafter, the mixture was incubated at 55 °C for another 20 min. Finally, the UV-Vis absorbance at 652 nm was measured, and three identical trials were conducted using three different OPs concentrations of 10, 50, and 100 ng/mL, respectively.

### 2.4. Quantitative Determination of OPs 

After discrimination of OPs, we repeated Section 2.3 to conduct a quantitative analysis of the six OPs using a variety of concentrations ranging from 10 ng/mL to 1000 ng/mL. Various fixations of OPs were added as target recognition substances to the response framework. The UV-Vis absorbance at 652 nm was estimated, and three identical trials were conducted. 

### 2.5. Specificity Analysis

By repeating Section 2.3, different interferences (PO_4_^2−^, Cu^2+^, Mg^2+^, Na^+^, Fe^3+^, Zn^2+^, Ca^2+^, Br^−^, K^+^, CO_3_^2−^, Cl^−^, and SO_4_^2−^) were added to the reaction system at a concentration of 100 ng/mL as target detection substances. The UV-Vis absorbance at 652 nm was estimated, and three identical trials were conducted. 

### 2.6. Discrimination of the OPs in Real Samples

Several fruits and vegetables were selected and tested using the proposed sensor for OPs recognition using Pa, Ma, Ro, and Pho. Samples of mangoes, ginger, broccoli, apples, tomatoes, spinach, bananas, potatoes, and cabbage, were bought locally from the nearest store. As utilized in the field, we reproduced the spraying phenomena of pesticides in food produce. After washing, each sample was separated into four equal parts, splashed with 150 ng/mL (Pa and Ma) and 200 ng/mL (Ro and Pho), and air-dried for 2 days before preparing the working solution as follows; 

One gram of roughly cut vegetables was weighed in a 10.0 mL rotator tube, and 5.0 mL of distilled water was added and shaken for 10 min and left to stand for a minute before being used for analysis. 

To obtain the working solution, the surface of the fruit was washed using 5.0 mL of refined water, and the gushing was the measured solution. We repeated Section 2.3 using the obtained samples and the UV-Vis absorbance at 652 nm was estimated. Three identical trials were conducted. 

## 3. Results

### 3.1. Characterization

Transmission electron microscopy (TEM) was utilized to study the structure of Pd@Pt as shown in Figure 1. Pd@Pt was synthesized with core-shell nanoparticles approach and exposed its concave on the surface [34]. Pd@Pt had an average size of 45 nm and a size distribution of 45 ± 5.0 nm. The structure of Au@Pd and the mesoporous Pd@Pt were further elaborated by N. Cheng et al. [38] and X. Wang et al. [34], respectively.

### 3.2. POD Activity Verification of Nanozymes

Previous research used various peroxidase substrates, such as TMB, o-phenylenediamine (OPD), and 2, 2’-azino-bis (3-ethylbenzothiazoline-6-sulfonic acid) (ABTS), as indicators for colorimetric reactions [50]. Due to its good color rendering, our study utilized TMB as a substrate for a redox reaction. In the presence of H_2_O_2_, nanozymes catalyzed the conversion of TMB to its oxidized form (oxTMB) and produced H_2_O. The TMB-H_2_O_2_ redox reaction mechanism is expressed below in Figure 2A. The activity of two nanozymes was evaluated, and Pd@Pt revealed a high effect on POD-like activity (Figure 2B,C). In the POD reaction, H_2_O_2_ reacts with nanozymes and produces reactive oxygen species such as hydroxyl radical (**^.^**OH). The produced **^.^**OH is accountable for further oxidation of the substrates [18], and then the effect of **^.^**OH on Pd@Pt POD-like activity was evaluated using ·OH scavengers. After the addition of ·OH scavengers (t-butanol and dimethyl sulfoxide (DMSO)) in the reaction systems, the Pd@Pt POD-like activity was altered (Figure 2D and Appendix A). This signifies that the presence of **^.^**OH is crucial for further oxidation of substrate. With this superior POD-like activity, Pd@Pt can be potentially used for latent applications and was selected for further experiments. The kinetic parameters of Pd@Pt were further studied, and the specific activity (SA) value was determined. 

### 3.3. Condition Optimization of Pd@Pt

Some interior or exterior factors may affect an enzyme’s reaction rate, including the amount of substrate, reaction pH, and temperature. An optimal temperature for many natural enzymes for humans is 37 °C [19], even though the optimal pH of enzymes depends on where they work. However, some enzymes function under various harsh conditions. For example, Pepsin performs best in gastric juice, which has an ideal pH of about 2 [51]. Although nanozymes are synthetic materials with enzyme mimic characteristics, they still reveal more stability even in harsh conditions that are even unbearable to natural enzymes [52]. Similar to natural peroxidase enzymes, POD-like nanozymes catalyze substrate in a pH- and temperature-dependent manner. Our study displayed the ideal pH and temperature of 4.6 and 45 °C, respectively (Figure 3A,B). The change in POD activity in pH and temperature ranges could be attributed to TMBs’ weak solubility and instability of H_2_O_2_ in neutral and alkaline conditions [53]. Although TMB is a good peroxidase indicator, it faces solubility and stability challenges in buffered solutions. When comparing our findings with those of horseradish peroxidase (HRP) [53], Pd@Pt had better thermal stability in a wide range of temperatures than HRP. In addition, we optimized the Pd@Pt concentration, and POD-like activity was directly proportional to the concentration (Figure 3C).

Regular enzymes are selective, but nanozymes can continue to work catalytically even under adverse environmental circumstances [18,52,53]. On that note, we examined the Pd@Pt catalytic activity under harsh acidic and alkaline conditions. After 8 h of reaction, the Pd@Pt revealed better stability and activity of greater than 85% (Figure 3D), which are far more stable than common enzymes and conventional enzyme mimics. This could be attributed to the synergistic property of individual metals due to their high lattice matching toward Pd@Pt alloys [34]. The persistence of nanozyme activity in adverse conditions was also reported by W. Yang, who elaborated the robustness of Prussian blue and other NPs with multi-enzyme activity [18]. In contrast to HRP [53], whose catalytic activity decreases when pH was below 5, Pd@Pt has better tolerance to harsh conditions. 

### 3.4. Catalytic Kinetics

Nanomaterials characteristically have reaction kinetics and mechanisms that are similar to those of ordinary enzymes. After confirming the POD-like activity of nanozymes, a nanozyme’s TMB time response analysis, correlation coefficients (R^2^), and kinetic parameters were further investigated. The TMB time response curve revealed a linear regression with R^2^ of approximately 1 (Figure 4A,B). Similar to natural enzymes, Pd@Pt followed the Michaelis–Menten kinetics and ping-pong mechanism [34]. The Michaelis–Menten method is among numerous and famous approaches utilized to assess the kinetics of enzymes whose conditions link the enzyme response rate (v) to the amount of substrate [*S*] and provide two key parameters (*V_max_* and *K_m_*). Generally, compared to biological enzymes, nanozymes have lower *K_m_* toward substrates TMB. The enzyme kinetics parameters (*V_max_* and *K_m_*) for Pd@Pt were 0.36002 Ms^−1^ and 0.26417 mM (Figure 4B) and 0.1823 Ms^−1^ and 0.28871 mM for Au@Pd (Figure 4C). Appendix A compares the kinetic parameters of several bimetallic NPs with HRP. A lower *K_m_* value shows better affinity linking of substrate and enzyme [50]. From the table, Pd@Pt NPs have lower *K_m_* which signify higher affinity for TMB than HRP. This may be due to the individual synergistic properties of Pd and Pt in the core-shell structure [34]. Specific activity (SA) of nanozymes was further studied to reveal a linear relationship between the catalytic activity and the number of nanozymes that catalytically produce 1 μmol of the product (Figure 4E,F). With these findings of Pd@Pt, potential studies can be developed.

### 3.5. Optimization 

AChE, ChOX, Pd@Pt nanozyme, and ACh concentration were optimized for better signal. As depicted in Appendix A, their solution’s absorbance increases as the concentration increases to its optimal amount. During optimization, the optimal values were 1 U/mL, 5 U/mL, 0.05 mg/mL, and 7 mg/mL, respectively. Unless otherwise specified, the above-optimized conditions served as the conditions of detection.

### 3.6. Sensitivity Analysis

As shown in Figure 1, we constructed a rapid colorimetric method for quantifying OPs using TMB as a substrate, a multi-enzyme cascade system, and Pd@Pt nanoparticles as catalysts. To stop the production of H_2_O_2_, Ca, Ma, Pa, Ph, Ro, and Pho were incubated at varying concentrations in an ACh/AChE/ChOX solution. Pd@Pt converted TMB to its oxidized form (oxTMB). Six OPs with concentrations of 10, 50, and 100 ng/mL were added to the reaction system to test the colorimetric sensor’s ability to distinguish them. The Pd@Pt responded differently to each OP since different OPs inhibited AChE in different ways. The solution’s color change was noticeable to the bare eye. Figure 5 elaborates on the successfulness of the sensor unit in quantifying the OPs by showing their distinct characteristics and relationships. To further validate the colorimetric sensor array’s recognition accuracy, the six OPs were distinguished using a hierarchical clustering technique. According to our results, the colorimetric sensor successfully identified all six OPs at each concentration (10 ng/mL (Figure 5A–C), 50 ng/mL (Figure 5D–F), and 100 ng/mL (Figure 5G–I)) and produced its unique response signals.

### 3.7. Quantitative Determination of OPs

A quantitative identification of Ca, Ma, Pa, Ph, Ro, and Pho was carried out by measuring their absorbance at various concentrations (10 ng/mL–1000 ng/mL). The results (Figure 6A–F) show that, there was a strong linear correlation between the OPs concentration and the absorbance value at 652 nm. The limit of detection (LOD) for Ca, Ph, and Ro was estimated to be 1 ng/mL (1 ppb), while the LOD for Ma, Pa, and Pho was 2.5 ng/mL (or 2.5 ppb). According to China’s national food safety standard for OPs [54,55], our results are lower than the set national limits. In addition, the six OPs had correlation coefficients (R^2^) greater than 0.93, with linear ranges of 10–200 ng/mL for Pa, Ro, and Pho and 25–200 ng/mL for Ca, Ma, and Ph, respectively. 

### 3.8. Specificity Analysis

According to our findings, a variety of OPs and carbamate pesticides were exposed by the constructed sensor. Each OP reacted differently, and a significant difference was exposed between Methomyl and other OPs (Figure 7A). This demonstrates that the constructed method can identify a variety of OPs. It is known that, when analyzing samples onsite, contaminating substances and OPs typically coincide. After incorporating these contaminants (100 ng/mL) into the system of 10 ng/mL OPs, the degree of colorimetric reaction was nearly identical to that of the blank group when contaminating substances were present (Figure 7B), indicating that the sensor array is unaffected by their presence and it has a certain anti-interference and feasibility in the detection of OPs.

### 3.9. Discrimination of the OPs in Real Samples

Following critical validation of the method, contents of Pa, Ma, Ro, and Pho were sprayed in various fruits and vegetables as explained in Section 2.6. After 48 h, the tested food samples were found to contain residual OPs, and more than 80% of the concentrations were recovered (Figure 8A,B). Taking Appendix A, Pa, as an example, it was significant that the color of actual samples could be clearly distinguished from the control sample with the naked eye. The spiked samples were light blue, while the control sample (without Pa) was dark blue. Additionally, compared to the others, the blue color of sample numbers 8 and 9 (ginger and mango) was the darkest and almost identical to the control sample (without Pa). The OP concentration in tomatoes, as determined by the UV signal, was higher than the LOD value of this colorimetric method but below the China standard guide [54]. Misuse of OPs has thus been a significant issue, and efficient supervision is required. Using this colorimetric method, OPs can be detected and strictly monitored.

## 4. Conclusions

OPs have become the preferred choice for insecticides in agriculture due to their high efficacy and easy accessibility. However, their improper use can negatively impact consumable products, the climate, and human well-being. Various methods like immunoassays and chromatographic techniques are used to identify OPs but face limitations, which necessitate a fast and straightforward approach. Therefore, a rapid colorimetric sensor was presented, which successfully quantified OPs in foods using TMB as a substrate indicator and a multi-enzyme cascade system. The method also utilized a bimetallic nanoparticle of Pd@Pt with a core-shell structure. Bimetallic NPs, especially double-metallic alloys of Pd and Pt, are known for their durability, high lattice matching, and multiple active sites, which are the potential factors for their enhanced synergistic properties. The proposed cascade method that utilizes Pd@Pt nanoparticles successfully quantified various pesticides, including Carbofuran, Malathion, Parathion, Phoxim, Rojor, and Phosmet, with a low detection limit range of 1–2.5 ng/mL, and short detection time compared to immunoassay and chromatographic methods. With a correlation coefficient (R^2^) value of over 0.93 and a linear range of 10–200 ng/mL, the proposed method recovered over 80% of residual OPs in food samples. Even though the proposed approach successfully quantified OPs at different concentrations, it poses a challenge to reaction time for cascade reactions. This gives researchers the need to develop a more rapid cascade colorimetric sensor (less than 50 min) for on-site quantification of OPs. Most of all, the method is simple and can be used for on-site detection without expensive equipment. Finally, Pd@Pt nanoparticle utilization in the bio-sensing field opens up a pothole for scholars to progress their research.

## Data Availability

The data used to support the findings of this study can be made available by the corresponding author upon request.

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
