# Peer review of "A Multi-Enzyme Cascade Response for the Colorimetric Recognition of Organophosphorus Pesticides Utilizing Core-Shell Pd@Pt Nanoparticles with High Peroxidase-like Activity"

_foods, 2023, doi:10.3390/foods12173319_

Round 1

Reviewer 1 Report

The article by Zainabu et al. discusses the development of a simple and rapid colorimetric sensor for the detection of organophosphorus pesticides (OPs). The proposed sensor is based on the peroxidase-like (POD) activity of Pt@Pd nanozyme and employs a cascade reaction system to distinguish between different OPs. Pt@Pd nanozymes, in particular, were not purified using centrifugation or dialysis, and the final solution contains all side products in addition to Pt@Pd. In this case, measuring POD or other physicochemical properties of Pt@Pd is extremely difficult. Furthermore, the experiments on the materials and the enzyme-like activities are not complete and not novel in the field of nanozyme research. As a result, I believe that this manuscript may not meet the requirements of the Journal. When the author revises the manuscript, the following other

1.        The introduction should include a specific research question or goal that the study intends to address. Why are two nanozymes, Pt@Pd and Au@Pd, used? 

2.        Authors should define Pt@Pd terminology: is it Pt decorated on Pd, Pt-Pd nanocomposites, or PtPd alloy? Similarly, Au@Pd. Explain crystallinity using XRD.

3.        The authors used DMSO to dilute TMB, and DMSO acts as an OH redical scavenger. Please elaborate on this.

4.        What is the solution concentration of Pt@Pd? For accurate concentration, an inductively coupled plasma atomic emission spectroscopy measurement is required.

5.        Because the Pt@Pd nanozymes were not purified, the final solution contains all side products in addition to Pt@Pd. It is extremely difficult to measure POD or other physicochemical properties of Pt@Pd in this scenario.

6.        Are the authors' ultrasonicating nanozymes solution before use, or do they aggregate and settle during storage?

7.        The authors used ascorbic acid as a surfactant, which is well known to inhibit the enzyme-like activity of Pt@Pd. Why did the authors use AA, and what about POD activity of Pt@Pd without AA?

8.        Explain why the catalytic activity of Pt@Pd changes with temperature and pH.

9.        Please correct the exponential fit curves in Figure 5.

10.    Compare the kinetic parameters estimated for nanozymes to catalyze the oxidation of TMB with HRP and previously reported nanozymes. A table is highly recommended.

Minor editing of English language required

Reviewer 2 Report

The authors report colorimetric detection of organophosphorus pesticides.  the manuscript is well-written and supported by detail experimental data. the discussion is comprehensive and supported by relevant references. Thus, I suggest that the manuscript need to do minor revisions. 

1. please provide a purity of the chemicals used in this work, in the part of materials and reagent

2. Line 98-107, please delete the numbering of the paragraph. follow the rule of writing in this journal. 

3. Fig.7: the figures should be edited, the inlet data is not clear for the readers. 

4. Fig.3 : what makes it possible for error bars to be really wide/big. is the provided data in the figures are the average of some datas in the same variations? it looks like that the deviations are big 

5. Does this detection suitable for any kind of organophosphorus perticides?

6. there are some writting errors, H2O2, "2" in this molecules should be subscripted. 

7. Inlet data in Fig.3 is not clear for readers. 

Reviewer 3 Report

The language used is of poor quality. I am not a native English speaker, but even for me, it is too difficult to comprehend. Some sentences are completely unreadable. I have pointed out some errors, but evidently, the entire text needs to be rewritten.

Line 20 – please, rephrase “the lead of simple excursion”. The sentence appears to be incorrect.

Fig. 8 – check “CO32–

Refs. 1 and 2 about nanozymes but not about pesticides. I think they should be cited in the corresponding paragraph

Line 40 – “their techniques” their?

Line 55 – “disagreeable circumstances”?

Line 57 – “high consistent” more stable?

Line 59 – “extraordinary recital”?

Line 70 – H2O2

Scheme 1 – please, decipher “Cho” in the figure caption

Line 84 – K2PtCl4 is not an acid

Line 85 – “polylactic acid nanoparticles (Pluronic F127)” Pluronic is a block polymer, not nanoparticles

Line 94 – “of the Czech Republic of the United States”???

Lines 91-96 – description of devices should be unified (see Instructions for Authors of Foods)

Line 97 – materials and methods again??

Line 99 – “of Pluronic F127 ultrasound”

Line 129 – range of acetate buffer is 3-5. Acetate buffer with pH 6.6, 7.6 and so on is a nonsense, because it does not have any buffering capacity.

Line 168 – “primary water”?

Line 169 – “sprayed with 150 ng/mL and 200 ng/mL” Concentrations of what?

Line 183 – “The nanozymes were synthesized with core-shell nanoparticles” what does it mean?

Line 184 – “middling size” average?

The language used is of poor quality. I am not a native English speaker, but even for me, it is too difficult to comprehend. Some sentences are completely unreadable. I have pointed out some errors, but evidently, the entire text needs to be rewritten.

Reviewer 4 Report

In this manuscript, authors describe " A Pt@Pd nanozyme with high peroxidase-like activity for colorimetric detection of organophosphorus pesticides via a multi-enzyme cascade reaction". Overall, this is a well-documented manuscript indicating good results. They described a simple and rapid colorimetric sensor that based on the high peroxidase-like (POD) activity of Pt@Pd nanozyme.  But there are some minor concerns to be addressed.

1. In the introduction part superority of nanostructures could be emphasized.

2. Error bars could be inserted for figures (Fig 4.C, fig S2….).

3. Conclusion part may be enlarged with more details.

4. Pls insert nanostructure based studies as a reference in the introduction part emphasizing the superorities such as;

- Yılmaz, G.E., Saylan, Y., Göktürk, I., Yılmaz, F., Denizli, A., 2022. Selective Amplification of Plasmonic Sensor Signal for Cortisol Detection Using Gold Nanoparticles. Biosensors 12, 482.. https://doi.org/10.3390/bios12070482

- Gokturk, I., Bakhshpour, M., Cimen, D., Yilmaz, F., Bereli, N., Denizli, A., 2022. SPR Signal Enhancement With Silver Nanoparticle-Assisted Plasmonic Sensor for Selective Adenosine Detection. IEEE Sensors Journal 22, 14862–14869.. https://doi.org/10.1109/jsen.2022.3186518

Round 2

Reviewer 1 Report

The authors' response about changes in enzyme-like properties with changes in temperature and pH is not convincing. I suggest going through this article https://doi.org/10.1002/admi.202101115. 

Reviewer 3 Report

The paper is very poorly written. The conclusions are not supported by the results and, in some cases, even contradict the findings. The characterization of nanoparticles is insufficient. After reading only three papers, I have identified multiple issues.

Why did authors compare Pd@Pt and Au@Pd nanoparticles, but not say pd and Pd@Pt or Pt and Pd@Pt? This choice is unclear and should be explained.

Line 174 - core-shell structure of nanoparticles is not supported by any results. Authors should present EDX mapping/cross-section (as in ref 34)

Line 178 - Au@Pd are not shown in Fig.1

Line 180 - "Au@Pd revealed nanopopcorn structure" - what is nanopopcorn? There is no TEM images of these nanoparticles in the main text and SI. How did the Authors measured their size? 

Line 187 - since no elemental analysis is presented, one cannot be sure that nanoparticles are bimetallic. Their structure should be characterized properly.

Line 183 - please, provide size distributions for both types of nanoparticles.

Line 197 - the Authors claim different mecghanisms of catalysis for both types of nanoparticles, but mechanism of Au@Pd activity is not studied at all.

Line 202, Fig. S1 - "t-butynol"? t-butanol?

Line 202 - In Section 2.2.2 the Authors describe TMB substrate, that already contains DMSO. However, in fig. 2D DMSO is used as a scavenger. TMB is hardly soluble at about 0.4 mg/ml (Section 2.2.2 and 2.2.3) in buffer with pH 4.6 without DMSO. PLease provide detailed protocol of scavengers study. Check presence of DMSO in substrate solutions in different sections of materials and methods (2.2.2, 2.2.3, 2.2.4, 2.2.5). In some experiments DMSO is added, in some experiments - is not added. Careful explanation is needed.

Lines 213-225 - more suitable for methods Section

Line 234 - "a pH of approximately 7 are optimal for many natural enzymes." Horseradish peroxidase, which is the most popular enzyme in in vitro assays is more active at acidic pH. The statement must be supported with references or removed/rephrased.

Line 244 - Authors claim that high pH of substrate can change mesoporous state of theitr catalyst and reference to paper 34. But in this paper non-mesioporous Pd@Pt was formed when pH was alkaline IN THE COURSE OF SYNTHESIS. I mean that there is no information about change in morfology of mesoporous Pd@Pt after their addition to alkaline conditions. Therefore the sentences "This was contributed to the change in Pd@Pt structure. At high pH and temperature, no uniform-size mesoporous structure were formed [34]. Only this facture is sufficient to affect the robustness of mesoporous Pd@Pt" should be removed.

Lines 237-244 - diplication of information from fig.4A and B. Should be removed. 

Line 248 - "discovered that the POD effect grew in proportion to the concentration." This information is self-evident and does not contribute to optimization. The mass activity of the nanozyme can be calculated from these data for comparative purposes. However, the simple statement that increasing the amount of catalyst leads to more product is trivial and should be removed.

Line 253 - which are far more stable than common enzymes and conventional enzyme mimics. This statement should be supported with reference or experiment. Enzymes are, indeed, sensitive to environment. But their mimics (including nanozymes) are usually much more stable. Specific examples are necessary for comparison.

Fig. 3 A and C. - results are confusing. In fig. 3A and Fig 2B I see almost NO activity of Au@Pd and very high activity of Pd@Pt. But from fig3C I see that mass activity of Pd@Pt is only 1.5 times higher? How is that possible? Results from Fig. 3 are not discussed in article at all. Three-panel figure and only one sentence in discussion??? (line 204). It seems like authors could not draw any useful information from these results.

In previous review report I asked Authors to rewrite their manuscript. Some issues have been fixed, but added paragraphs and sentences are confusing.

Line 200 - scavengers which will remove the radicals and alter the Pd@Pt nanozyme activity and alter the POD-like activity of the nanozyme. These are the same.

Line 214 -225 - very strange pharses. In previuos review I asked Authors to check English, but some more unclear sentences have been added: "synergist movement of the unit nanozyme" WHAT?, "μmol of the produce" PRODUCT?, "articulated per milligram" CONVERTED?, "autocatalysis" WHAT?, "determining the  gradient" SLOPE?

Line 227 - "-time response bends", response-time bends' CURVES? 

Line 245 - ". Only this facture" FACTURE???

Line 249 - "Due to their extreme nature, regular enzymes" What is EXTREME NATURE?

Line 252 - function? ACTIVITY?
